# Mutual Effort for Efficiency: A Similarity-based Token Pruning for Vision Transformers in Self-Supervised Learning

**Sheng Li**[*1]  **Qitao Tan**[*2]  **Yue Dai**[1]  **Zhenglun Kong**[3]  **Tianyu Wang**[1]
**Jun Liu**[4]  **Ao Li**[5]  **Ninghao Liu**[2]  **Yufei Ding**[6]  **Xulong Tang**[1]  **Geng Yuan**[2]
[1]University of Pittsburgh  [2]University of Georgia  [3]Harvard University
[4]Northeastern University  [5]University of Arizona  [6]University of California, San Diego
{shl188,xulongtang}@pitt.edu          {qitaotan,geng.yuan}@uga.edu

## Abstract

Self-supervised learning (SSL) offers a compelling solution to the challenge of extensive labeled data requirements in traditional supervised learning. With the proven success of Vision Transformers (ViTs) in supervised tasks, there is increasing interest in adapting them for SSL frameworks. However, the high computational demands of SSL pose substantial challenges, particularly on resource-limited platforms like edge devices, despite its ability to achieve high accuracy without labeled data. Recent studies in supervised learning have shown that token pruning can reduce training costs by removing less informative tokens without compromising accuracy. However, SSL's dual-branch encoders make traditional single-branch pruning strategies less effective, as they fail to account for the critical cross-branch similarity information, leading to reduced accuracy in SSL. To this end, we introduce SimPrune, a novel token pruning strategy designed for ViTs in SSL. SimPrune leverages cross-branch similarity information to efficiently prune tokens, retaining essential semantic information across dual branches. Additionally, we incorporate a difficulty-aware pruning strategy to further enhance SimPrune's effectiveness. Experimental results show that our proposed approach effectively reduces training computation while maintaining accuracy. Specifically, our approach offers 24% savings in training costs compared to SSL baseline, without sacrificing accuracy.

## 1 Introduction

Self-supervised learning (SSL) (Chen et al., 2020; Li et al., 2024a) is a training paradigm that addresses the need for extensive labeled data in traditional supervised learning (Yuan et al., 2022; Ollivier et al., 2022; Li et al., 2022a). Due to the success of Vision Transformer (ViT) in supervised learning, researchers have attempted to integrate ViT into SSL frameworks (Caron et al., 2021; Chen et al., 2021). Masked Image Modeling (MIM) and discriminative methods are two prominent SSL paradigms for ViT. MIM-based methods, such as BEiT (Bao et al., 2021), MAE (He et al., 2022), and SimMIM (Xie et al., 2022), reconstruct masked portions of input images, enabling models to capture fine-grained image details. This makes them well-suited for tasks like object detection and segmentation, which benefit from detailed local features. In contrast, discriminative SSL methods focus on aligning representations from differently augmented views of the same image, prioritizing global representation learning (Caron et al., 2021; Zheng et al., 2021; Cao et al., 2023). These methods excel in classification tasks, achieving higher accuracy ($\sim$3%) on ImageNet classification compared to MIM-based approaches (Oquab et al., 2023). In this paper, we focus on discriminative self-supervised learning, an important and promising area in self-supervised learning research.

Despite SSL achieving high accuracy without the necessity to use labeled data, its high training costs, which require more than $8\times$ the training iterations and more than $2\times$ the computational cost per iteration compared to supervised learning (Wen & Li, 2021), remain a significant barrier. This is particularly challenging for resource-limited platforms, such as edge devices, urging the need for more

---

*Equal contribution.

efficient SSL approaches to enable practical ViT deployments. Recent studies have highlighted the effectiveness of token pruning in reducing the training costs of Vision Transformers in conventional supervised learning contexts (Liang et al., 2022b; Rao et al., 2021; Kong et al., 2022). These methods selectively eliminate less informative tokens during training, thereby decreasing computational demands and enhancing training efficiency. For example, certain approaches successfully reduced computational costs by 24% in ViT models without compromising accuracy (Liang et al., 2022b). However, the training paradigm of discriminative SSL differs significantly from supervised learning. It utilizes dual-branch Siamese encoders, each processing distinct augmented views of the same input, aiming to maximize the similarity between representations of these different views. In contrast, supervised learning trains the model using a single branch and is guided by explicit labels. As such, it is natural to raise a question: *Do existing token pruning strategies remain effective under the SSL paradigm?*

To address this inquiry, we apply a representative token pruning method that prunes tokens with lower attention scores (Liang et al., 2022b; Xu et al., 2022; Song et al., 2022) to SSL and evaluate its effectiveness. However, this method results in a considerable drop in accuracy, suggesting that existing token pruning strategies may not effectively enhance SSL efficiencies. This is because such an approach only evaluates token importance based on self-attention scores within a single branch, does not consider the crucial cross-branch similarity information that SSL relies on, and therefore may remove features that are critical to SSL performance.

Fortunately, the augmented input images and feature maps on the two branches of a Siamese network inherently share a certain degree of similarity. For example, although differently augmented, divergent views from the same image typically represent the same objects. This natural similarity presents significant opportunities for enhancing token pruning and training. Firstly, by leveraging the inherent similarities across branches, we can pair tokens from the two branches and prune them in pairs, maintaining essential cross-branch semantic consistency. Additionally, inspired by previous research (Zheng et al., 2019; Tan et al., 2023; Bian et al., 2021; Ma et al., 2022) on improving model performance by adjusting training difficulty, selectively pruning token pairs with varying degrees of similarity allows us to strategically control the difficulty of the SSL process, which further improves the model accuracy.

Building on the insights above, we propose SimPrune, a token pruning approach for Vision Transformers in discriminative self-supervised learning, which leverages the unique characteristics of SSL to enhance training efficiency. Our approach consists of two key components: First, we introduce a similarity-based token pruning method that uses cross-branch similarity information to pair and prune tokens, which allows us to eliminate non-essential tokens without compromising SSL performance. Second, we introduce a difficulty-aware pruning strategy that strategically prunes token pairs with different similarity levels at different training stages, further boosting the model's final performance. Our approach effectively utilizes mutual information contributed by dual branches to boost training efficiency (i.e., Mutual Effort for Efficiency). In summary, our major contributions are as follows:

- We conduct a preliminary study to analyze the effectiveness of conventional single-branch token pruning on discriminative self-supervised learning. We find that conventional token pruning is not desirable for SSL because it overlooks the critical cross-branch similarity information necessary for effective token pruning.

- We propose a novel token pruning approach, SimPrune, tailored specifically for SSL. This method utilizes cross-branch similarity to guide token pruning, maintaining crucial semantic consistency across different augmented views of the same image, thus enhancing model efficiency without compromising accuracy.

- We introduce a difficulty-aware pruning strategy to enhance our token pruning approach, which prunes token pairs at different similarity levels throughout different training stages. This strategy offers an effective and efficient way to control the training difficulty, further optimizing the model performance.

- We conduct a comprehensive evaluation of our proposed SimPrune approach. Compared to the SSL baseline, SimPrune reduces computation costs by 24% without compromising accuracy. Additionally, when compared to the popular token pruning method EViT and the token merging method ToMe, SimPrune outperforms them by achieving 3.1% and 3.2% higher accuracy, respectively, while consuming similar training costs.

## 2 BACKGROUND AND RELATED WORKS

### 2.1 DISCRIMINATIVE SELF-SUPERVISED LEARNING FOR VISION TRANSFORMERS

Self-supervised learning (Chen et al., 2020; Grill et al., 2020) offers a compelling alternative to traditional supervised learning by eliminating the need for extensive labeled data (Li & Feng, 2019; Li et al., 2019; Hou et al., 2022; Li et al., 2022b; 2024c). This method exploits inherent patterns within the data itself to train models effectively. Given the success of Vision Transformers in supervised tasks, researchers have explored their integration into SSL frameworks (Caron et al., 2021; Chen et al., 2021). Considering that extracting detailed, context-rich representations from unlabeled data is essential in SSL, ViTs' proficiency in capturing complex dependencies in data makes them particularly effective for SSL.

DINO (Caron et al., 2021) is a prominent discriminative SSL framework tailored for ViT. It employs two branches: an online (i.e., student) branch and a target (i.e., teacher) branch, each processing different augmented views of the same image. DINO's learning process is based on the principle of self-distillation, where the online encoder is trained to replicate the output of the target encoder. The framework operates by maximizing the similarity between the outputs from these two branches. The two branches use identical model architectures, with the online encoder being updated via back-propagation for immediate learning adjustments, while the target encoder is updated by a momentum-based approach (He et al., 2020). DINO offers the option to enhance its performance by employing local views (smaller cropped patches). Specifically, the online branch processes both global and local views, while the target branch processes only global views. The outputs of both view types in the online branch are then aligned with the global view output from the target branch. Incorporating local views helps facilitate the learning of rich and granular features.

MoCo v3 (Chen et al., 2021) adapts the Momentum Contrast approach for ViT. It not only maximizes the similarity between representations of differently augmented views of the same image but also concurrently reduces the similarity with representations of other images in the batch. However, despite the advantages, self-supervised learning with ViT is challenged by significant computation costs and slow convergence speeds (Wen & Li, 2021), which limits their effectiveness and practicality.

### 2.2 SPARSE VISION TRANSFORMER

Token pruning is a promising technique for reducing the computation costs for ViTs. The most popular strategy to select the tokens to be pruned is utilizing attention scores to determine the importance of each token, which is adopted by many token pruning approaches (Liang et al., 2022b; Kong et al., 2022; Xu et al., 2022; Song et al., 2022; Fayyaz et al., 2022; Yu & Wu, 2023). Tokens that receive lower attention scores are often considered less critical and are thus candidates for pruning. In addition to the attention-based strategy, there are other strategies to determine which tokens to prune. For example, some models incorporate a trainable component that explicitly scores each token's relevance to the task (Rao et al., 2021; 2023). This scoring function is optimized during training to allow the model to learn task-dependent criteria for pruning.

Token merging, as a complementary approach to pruning, has been gaining traction for its ability to condense information without significant losses. A recent work (Bolya et al., 2023) proposes to merge similar tokens using a lightweight matching algorithm, effectively condensing information and reducing the number of tokens processed. There are also other sparse ViT designs beyond token-level sparsity, such as sparse attention mechanisms (Liu et al., 2021; Kong et al., 2023; Li et al., 2022c). However, token reduction methods are particularly favored due to their simple implementation and outstanding performance in computation reduction.

## 3 DOES EXISTING TOKEN PRUNING WORK WELL FOR SSL?

In supervised learning, there is only one single branch to process input data and the learning process is guided by explicit labels. Differently, in discriminative self-supervised learning, the Siamese encoders are generally used in the two branches, where each branch processes distinct augmented views from the same input image. And the model (i.e., encoder) is trained by maximizing the similarity between representations of differently augmented views of the same image. Considering that the training

paradigm of SSL diverges significantly from conventional supervised learning, it is natural to question whether existing token pruning methods are truly desirable for SSL. Therefore, we first set out to investigate this question. Here, we select one of the most representative attention-based token pruning approaches (Liang et al., 2022b) for our investigation, applying it to SSL to assess its performance.

## 3.1 REVISITING ATTENTION-BASED TOKEN PRUNING IN SUPERVISED LEARNING

The attention-based pruning approach is the most popular token pruning approach and has been used by many previous token pruning works in supervised learning, as discussed in Section 2, due to its outstanding performance, simplicity, and ease of implementation, The key idea of this type of approach is to use the self-attention scores to evaluate the importance of the tokens and prune those less important tokens. Specifically, Vision Transformers employ a multi-head self-attention (MHSA) mechanism to process a sequence of input image tokens (i.e., image patches). Each token is first linearly transformed into matrices of queries ($Q$), keys ($K$), and values ($V$). Then the attention operation is performed as:

$$\text{Attention}(Q, K, V) = \text{Softmax}\left(QK^T/\sqrt{d}\right)V. \tag{1}$$

Here, $d$ represents the dimensionality of the keys and it is used to scale the outputs to stabilize the gradients during training. Then the attention scores ($Softmax(QK^T/\sqrt{d})$) are used to multiply the values ($V$) to produce the output, effectively synthesizing the information across different tokens. As such, tokens with higher attention scores have more significant contribution to the final output.

## 3.2 APPLYING EXISTING TOKEN PRUNING APPROACH TO SSL

In this experiment, we follow the implementation of a recognized paper EViT (Liang et al., 2022b), which applies the representative attention scores from the [CLS] token to other tokens to identify the most important tokens. Specifically, we keep the top-k (k is a hyperparameter) tokens with the highest attention scores from the [CLS] token and prune the non-topk tokens. The reason why we use the attention of the [CLS] token to other tokens is that the [CLS] token is designed to capture the global contextual information of the entire input image, making it ideal for a holistic understanding of the input image. Following the setting in previous works (Liang et al., 2022b; Kong et al., 2023), we apply token pruning only at certain transformer blocks (i.e., the 4th, 7th, and 10th transformer blocks in DeiT). Once a token is pruned at a certain layer, it will not appear in subsequent layers.

Table 1: Results of applying attention-based token pruning approach to the self-supervised learning on ViT. DINO is used as the training framework in this experiment (w/o local views). The encoder is DeiT-S and it is trained on the ImageNet-1k for 300 epochs. The accuracy results are obtained through linear evaluation. kr represents the token keep rate.

| Method | Training FLOPs | SSL Acc. | $\Delta$Acc. (SSL) | $\Delta$Acc. (SL) |
|---|---|---|---|---|
| SSL w/o token prune (baseline) | 100% | 57.16$\pm$0.22 | – | – |
| Atten-based token prune (kr = 0.9) | 87% | 56.65$\pm$0.26 | -0.51 | -0.10 |
| Atten-based token prune (kr = 0.8) | 76% | 54.68$\pm$0.30 | -2.48 | -0.21 |

Table 1 shows the experimental results. We use DeiT-Small (Touvron et al., 2021) as the encoder and use ImageNet dataset (Deng et al., 2009) as an example. We use DINO as the training framework and directly apply the attention-based pruning to the encoders of two branches. We do not incorporate the local views in this preliminary experiment (we include experiments that incorporate local views later in Section 5). The keep rate in the table indicates the ratio of retained tokens after pruning to the total number of tokens before pruning. For example, if we set the keep rate to 0.8 and apply token pruning at the 4th, 7th, and 10th blocks, 80% of the tokens are retained after the 4th block. Subsequently, due to the cumulative pruning effect, only 64% of the original tokens remain after the 7th block. The column $\Delta Acc.(SSL)$ shows the accuracy difference compared to the baseline model under the SSL setting, while the $\Delta Acc.(SL)$ shows the accuracy difference under the original supervised learning setting counterparts using the same network and dataset. We select two relatively high keep rates (0.9 and 0.8) for this experiment. As shown in the table, these two keep rate configurations incur only marginal accuracy loss (about 0.2%) in the supervised learning setting. However, there is a significant

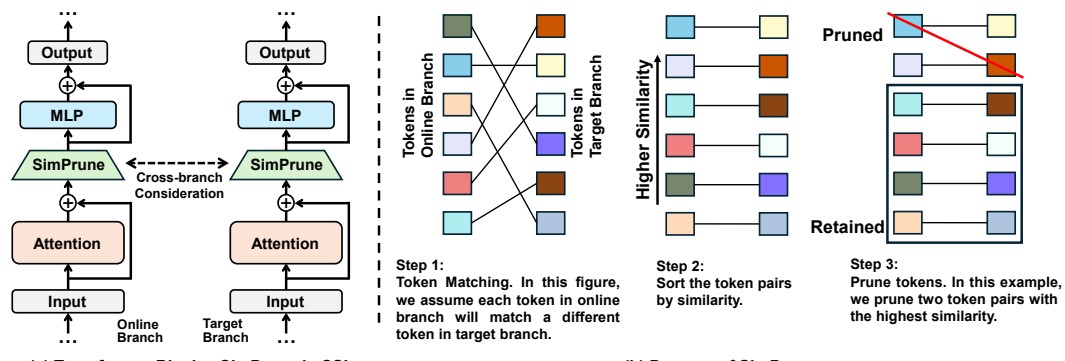

Figure 1: Overview of SimPrune. (a) SimPrune can be directly applied to a transformer block in self-supervised learning. (b) The process of SimPrune. We assume each token in the online branch matches a different token in the target branch in this example. And we present the case that more than one tokens in the online branch match a token in the target branch later in Figure 2.

accuracy drop of more than 2.4% in SSL with a keep rate of 0.8. These results show that the SSL is more sensitive (severe accuracy drop) to attention-based token pruning, indicating conventional single-branch attention-based pruning may not be suitable for SSL.

The limitations of applying the popular attention-based token pruning approach to SSL stem from the differences between the training paradigm of SSL and supervised learning. In SSL, models are trained by maximizing the similarity of representations from different augmented views of the same source image, relying on cross-branch similarity information (Li et al., 2024b; Misra & Maaten, 2020; Reed et al., 2021; Zbontar et al., 2021). In contrast, token pruning approaches developed for supervised learning typically evaluate token importance based solely on attention scores from a single branch. They often remove tokens with lower self-attention scores—usually associated with background features (Liang et al., 2022b). Such an approach does not consider the crucial cross-branch information that is essential for SSL, potentially leading to the removal of features critical for SSL performance. For example, asymmetric pruning may occur where tokens containing the same semantic information across two branches are pruned inconsistently, causing a mismatch in the representations and potentially impairing the model's ability to learn robust features. We use attention-based pruning as an example and provide a visualization for this issue in Figure 4 and Section 5.5. It is also important to note that this issue is not exclusive to the attention-based token pruning approach but also exists in all token reduction approaches that overlook cross-branch information, such as the popular Token Merging approach ToMe. A more detailed comparison of these approaches is presented in the evaluation section (Section 5.2).

## 4 SIMPRUNE DESIGN

In this section, we propose a similarity-based token pruning approach, namely SimPrune, which leverages the inherent similarity information from both two branches to guide the token pruning in self-supervised learning. In addition, we propose applying a difficulty-aware pruning strategy to our pruning approach. The overview of SimPrune is shown in Figure 1. While we follow the prior token pruning works (Liang et al., 2022b; Kong et al., 2023) that prune tokens only at certain transformer blocks (e.g., the 4th, 7th, and 10th blocks) in this paper, it is important to note that SimPrune can be applied at any transformer block, as shown in Figure 1a.

### 4.1 LEVERAGING CROSS BRANCH SIMILARITY FOR TOKEN PRUNING

Suppose we perform token pruning at the $i^{th}$ transformer block, the pruning process of SimPrune is as follows (visualized in Figure 1b). We first conduct the token matching for the image tokens in the online and target encoders in layer $i$ (step 1 in Figure 1b). Specifically, for each token (excluding the [CLS] token) in the online encoder, we match it to its most similar tokens in the target encoder. Here, the similarity is quantified using the cosine similarity. After the token matching, we would have $N$ token pairs across two branches, where $N$ is the number of image tokens in the online branch. We

Table 2: Evaluation of two static and two dynamic strategies of cross-branch similarity-based token pruning. DINO (w/o local views) is used as the training framework. The encoder is DeiT-S and it is trained on ImageNet-1k for 300 epochs. Toke pruning is performed at the 4th, 7th, and 10th transformer blocks. The keep rate for all methods is 0.8.

| Method category | Method | Accuracy |
|---|---|---|
| — | SSL without token pruning (baseline) | 57.16±0.22 |
| | Attention-based token pruning (keep rate = 0.8) | 54.68±0.30 |
| Static Similarity-based | Prune most similar pairs | 56.41±0.16 |
| | Prune most dissimilar pairs | 56.07±0.32 |
| Dynamic Similarity-based | Prune most dissimilar pairs at first half of epochs, and prune most similar pairs at second half of epochs | 56.90±0.21 |
| | Prune most similar pairs at first half of epochs, and prune most dissimilar pairs at second half of epochs | 55.76±0.19 |
| Sliding window Similarity-based (Ours) | Prune most dissimilar pairs at the beginning, gradually shift to pruning more similar pairs | 57.20±0.28 |

match each token in the online branch to its most similar counterpart in the target branch so that we can ensure semantic consistency for the paired tokens. Pruning at the granularity of these token pairs guarantees that the representations in both branches maintain consistent semantic information.

Next, we sort the token pairs by their similarity value (step 2 in Figure 1b). The key question then arises: which token pairs should be pruned in SSL? Generally, there are two potential strategies for pruning: i) prune the most similar pairs or ii) the most dissimilar pairs. Intuitively, since SSL trains the model by maximizing the similarity of the outputs from its two branches, pruning the most similar pairs and retaining those with lower similarity could increase the training difficulty for the model. This might enhance its ability to learn more robust and generalized features, as it faces greater challenges in recognizing connections between less similar views (Tian et al., 2020). On the contrary, pruning the most dissimilar pairs allows the model to focus on more similar parts, which may reduce the complexity of the learning process. We conduct a preliminary study to evaluate the effectiveness of these two strategies.

The results are presented in Table 2 and these two strategies are categorized as "static" in the table as they always prune the most similar/dissimilar token pairs during the whole training process. As one can observe, our static cross-branch similarity-based token pruning methods generally yield higher accuracy than the attention-based method, with an improvement of more than 1.7% when pruning the most similar token pairs. However, there remains a 0.75% accuracy drop compared to the baseline without pruning, suggesting that further optimization is needed. Since the model's capabilities evolve throughout training, different training stages may require varying levels of training difficulty (Bengio et al., 2009; Soviany et al., 2022), necessitating the use of different pruning strategies. To explore this, we set to investigate a dynamic pruning strategy in the following Section 4.2.

**Addressing imbalanced token pruning in two branches.** As depicted in Figure 2, during the token matching process, more than one token in the online branch may match the same token in the target branch. In the example in Figure 2, when pruning token pairs based on similarity, it is possible to prune the two token pairs with the many-to-one issue. It causes an imbalance problem, where the number of tokens pruned in the target branch is less than online branch. In our practice, we find that the number of tokens pruned in the two branches generally differs by less than 10%. To address this issue, we propose using an attention-based pruning approach to further prune tokens in the target branch, ensuring that the number of tokens in both branches remains the same and meets the predefined pruning ratio. Note that this method is already applied in the experiments in Table 2 for a fair comparison.

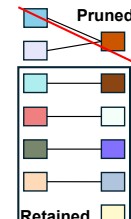

Figure 2: Many-to-one matching.

## 4.2 DIFFICULTY-AWARE PRUNING STRATEGY

Some studies have demonstrated that gradual adjustment of the difficulty of training tasks during the self-supervised learning process can effectively improve the model's performance and robust-

ness (Zheng et al., 2019; Tan et al., 2023; Bian et al., 2021; Ma et al., 2022). Specifically, these works suggest starting with simpler tasks for models with limited capabilities in the early training stage, and gradually increasing the training difficulty in later training stages as the model's capabilities evolve.

Based on these findings, we conduct a preliminary experiment to explore whether adjusting task difficulty can also yield benefits in self-supervised learning. In this preliminary study, we simply divide the whole training epochs into two halves, with the first half of epochs as the early training stage and the second half as the late training stage. We apply different pruning strategies in these two stages and evaluate their effectiveness. The results are shown in Table 2 and these strategies are categorized as "dynamic" in the table. As one can observe, the strategy that prunes the most dissimilar token pairs (i.e., retain similar tokens) at the early training stage while prunes the most similar token pairs (i.e., retain dissimilar tokens) at late training stages yields better accuracy (56.90%) than the above-mentioned static pruning methods, with 0.49% higher than the pruning strategy that always prunes the most similar token pairs. These results demonstrate that SSL also prefers reducing training complexity in the early stages and increasing the difficulty in the later stages.

As such, we propose applying this difficulty-aware pruning strategy to our cross-branch similarity-based token pruning approach. Considering that the model's capability evolves progressively during the training process, we aim to gradually adjust the training difficulty rather than arbitrarily dividing the training process into several fixed stages. This dynamic and gradual adjustment is achieved using a sliding window mechanism based on token similarity values. All the tokens within the window are retained after pruning, and the window size is determined by a pre-defined keep rate. This mechanism is illustrated in Figure 3. Initially,

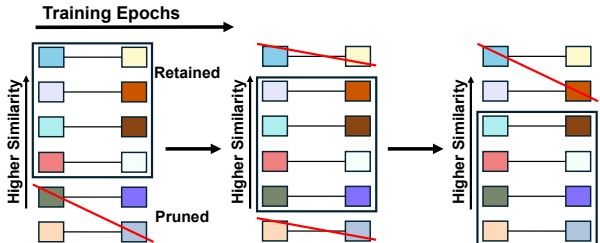

Figure 3: Illustration of sliding window mechanism. The black box indicates the window and the tokens inside are retained after pruning. It shows the pruning strategy in three different epochs.

the window includes token pairs with the highest similarity (i.e., pruning token pairs with the lowest similarity), making the task easier during the early training stages. As training proceeds, the window moves linearly with the training epochs to include token pairs with lower similarity (i.e., prune more token pairs with higher similarity), gradually increasing the task difficulty. By the end of the training, the window eventually incorporates the most dissimilar pairs. By integrating this difficulty-aware pruning strategy into our cross-branch similarity-based token pruning approach, we offer an effective and efficient way to control task difficulty without incurring additional computation overhead. As shown in Table 2, applying the sliding window mechanism to our pruning approach achieves the highest accuracy (57.20%) among all methods, highlighting its effectiveness.

## 5 EVALUATION

### 5.1 EXPERIMENTAL SETUP

In this section, we evaluate the performance of our proposed token pruning approach SimPrune. We use five representative datasets: ImageNet-1k, CIFAR-10, CIFAR-100 (Krizhevsky & Hinton, 2009), along with two fine-grained datasets Stanford Cars (Krause et al., 2013) and FGVC Aircraft (Maji et al., 2013). We use the DeiT model as the encoder. The batch size in the experiments is set to 128, the learning rate is 0.001, and we adopt the half-precision FP16 during training for efficiency. We also provide the results of using full-precision FP32 and a larger batch size of 1024 in Appendix A.5. In our experiment, we perform the token pruning at the 4th, 7th, and 10th transformer blocks of DeiT for all the evaluated approaches, following the setting in previous work (Liang et al., 2022b; Kong et al., 2023). The overhead introduced by SimPrune is included in the reported training FLOPs and time results and we present a detailed overhead analysis in Appendix B. We compare our proposed SimPrune with two other representative token reduction approaches: attention-based token pruning method EViT (Liang et al., 2022b) and token merging method ToMe (Bolya et al., 2023). In addition, we also evaluate the compatibility of SimPrune with two recent efficient self-supervised

Table 3: Comparison of different methods. DINO is used as the training framework. The encoders are DeiT-T, DeiT-S, and DeiT-B, which are trained on ImageNet-1k for 300 epochs.

| Keep Rate | Method | DeiT-T | | | DeiT-S | | | DeiT-B | | |
|---|---|---|---|---|---|---|---|---|---|---|
| | | Accuracy | Training FLOPs | Training Time | Accuracy | Training FLOPs | Training Time | Accuracy | Training FLOPs | Training Time |
| – | DINO | 55.71±0.23 | 100% | 100% | 62.49±0.21 | 100% | 100% | 64.56±0.15 | 100% | 100% |
| 0.9 | EViT | 55.24±0.24 | 87% | 93% | 62.25±0.36 | 87% | 95% | 64.35±0.16 | 87% | 95% |
| | ToMe | 55.38±0.17 | 87% | 94% | 61.92±0.20 | 87% | 94% | 64.12±0.13 | 87% | 95% |
| | SimPrune | 55.79±0.22 | 87% | 93% | 62.39±0.31 | 87% | 94% | 64.50±0.28 | 87% | 95% |
| 0.8 | EViT | 53.13±0.19 | 76% | 88% | 60.06±0.29 | 76% | 87% | 62.27±0.29 | 76% | 89% |
| | ToMe | 53.65±0.27 | 76% | 86% | 60.34±0.22 | 76% | 87% | 61.58±0.31 | 76% | 88% |
| | SimPrune | 55.66±0.30 | 76% | 86% | 62.31±0.11 | 76% | 85% | 64.21±0.27 | 76% | 89% |
| 0.7 | EViT | 52.30±0.41 | 65% | 78% | 58.40±0.24 | 65% | 75% | 61.12±0.14 | 65% | 82% |
| | ToMe | 52.01±0.35 | 65% | 77% | 58.68±0.38 | 65% | 78% | 60.71±0.25 | 65% | 80% |
| | SimPrune | 55.38±0.25 | 65% | 77% | 61.90±0.32 | 65% | 76% | 63.85±0.44 | 65% | 79% |

Table 4: Experiments on fine-grained image datasets. The encoder is DeiT-S and it is trained for 100 epochs. DINO is used as the training framework. The keep rate for token pruning is set to 0.8.

| Method | Stanford Cars | | | FGVC Aircraft | | |
|---|---|---|---|---|---|---|
| | Accuracy | Training FLOPs | Training Time | Accuracy | Training FLOPs | Training Time |
| DINO | 52.87±0.12 | 100% | 100% | 55.70±0.16 | 100% | 100% |
| EViT | 50.72±0.28 | 76% | 83% | 53.81±0.17 | 76% | 86% |
| ToMe | 49.56±0.18 | 76% | 87% | 53.15±0.24 | 76% | 85% |
| SimPrune | 52.61±0.24 | 76% | 84% | 55.48±0.26 | 76% | 86% |

learning methods. We follow the linear evaluation protocol (Goyal et al., 2019) in our experiments for evaluation, where the pre-trained model is fixed, and only the appended linear classification layer undergoes fine-tuning. We run the experiments on an NVIDIA A100 GPU. All results in this paper are the average of three runs with different random seeds.

## 5.2 MAIN RESULTS

**Experiments on ImageNet.** Table 3 presents the experimental results of different methods. Here we utilize DINO as the training framework and set the number of local views to six as the setup in (Caron et al., 2021). Details on how we handle local views in our experiments are provided in Appendix C. Compared to the DINO baseline, our proposed SimPrune approach greatly reduces training costs with minimal impact on accuracy. For instance, at a keep rate of 0.8, SimPrune reduces training FLOPs by 24% and training time by 13% on average, with less than 0.2% accuracy drop. In addition to computation cost reduction, SimPrune can also benefit memory usage during training, as memory usage is primarily associated with storing activations and gradients during the forward and backward propagation. By setting the keep rate to 0.8, SimPrune can achieve 11% memory usage reduction.

When compared to the attention-based token pruning method EViT, SimPrune consistently yields significantly higher accuracy with similar training time and FLOPs at the same keep rates. Notably, with a keep rate of 0.7, SimPrune provides 3.1% higher accuracy on average. In addition to EViT, we also compare SimPrune against the token merging approach ToMe. ToMe merges similar tokens within each branch to condense information and reduce the number of tokens for subsequent layers. Consistent with our token pruning experiments, token merging is performed at the same transformer blocks—specifically, the 4th, 7th, and 10th blocks. As shown in the table, at the same keep rates, SimPrune achieves much higher accuracy than ToMe while consuming comparable training time and training FLOPs. For instance, SimPrune delivers 3.2% higher accuracy than ToMe with a keep rate of 0.7, demonstrating the superiority of our proposed SimPrune approach.

SimPrune's superior performance stems from its use of similarity information across two branches to guide token pruning. In contrast, both EViT and ToMe rely on single-branch information (e.g., single-branch attention scores and single-branch token similarity information) and overlook the crucial cross-branch similarity information, which is vital for self-supervised learning.

**Experiments on fine-grained image datasets.** We further evaluate our proposed SimPrune using two fine-grained datasets, Stanford Cars and FGVC Aircraft. In this experiment, we follow the setup used in prior work (Shu et al., 2023). The encoder DeiT-S is initialized using ImageNet-pretrained weights. Then the DeiT-S is trained on the Cars and Aircraft datasets using self-supervised learning, after which the model is evaluated using the linear evaluation. As shown in Table 4, SimPrune delivers a 24% reduction in computation FLOPs and a 15% reduction in training time compared to the DINO baseline on average, while incurring only a marginal accuracy loss of 0.2%. Moreover, SimPrune outperforms both EViT and ToMe, achieving 1.8% and 2.7% higher accuracy, respectively, with comparable training time and FLOPs. These results demonstrate the effectiveness of SimPrune in processing fine-grained images.

For a more comprehensive evaluation, we also extend our SimPrune with a dynamic pruning strategy and compare SimPrune to more token reduction methods. These results are shown in Appendix A.1. we also conduct experiments on CIFAR dataset (Appendix A.2) and two other SSL framework MoCov3 (Chen et al., 2021) (Appendix A.3) and DINOv2 (Oquab et al., 2023) (Appendix A.4).

## 5.3 COMPATIBILITY OF SIMPRUNE WITH OTHER EFFICIENT SSL METHODS

In this section, we evaluate the compatibility of SimPrune with two recent efficient self-supervised learning methods (Addepalli et al., 2022; Koçyiğit et al., 2023). These two works are complementary to our proposed SimPrune. Specifically, Koçyiğit et al. (2023) introduces a combination of efficient training strategies (short as ETS) to accelerate model convergence by utilizing an innovative learning rate schedule, an adaptive input image resolution schedule, and a novel augmentation technique aimed at improving the quality of augmented images. Addepalli et al. (2022) identifies the noise present in the training objective as a key factor to the slow convergence in SSL. To address this, they propose rotation prediction (termed Rotation) as an additional, noise-free training objective, helping to expedite model convergence.

On the other hand, SimPrune operates in a different dimension, which removes less important tokens during training. To ensure a fair comparison, we adjust training epochs for these two efficient SSL methods to match the accuracy of baseline DINO. Then, we integrated SimPrune with these two methods. Table 5 shows that combining SimPrune with the ETS and Rotation can further reduce the training cost by 17% and 15% without compromising accuracy, respectively, demonstrating the compatibility and effectiveness of SimPrune.

Table 5: Compatibility of SimPrune. The dataset is ImageNet and the encoder is DeiT-S.

| Method | Accuracy | Training Time |
|---|---|---|
| DINO (baseline) | 62.49±0.21 | 100% |
| ETS | 62.56±0.36 | 71% |
| Rotation | 62.45±0.25 | 64% |
| ETS + SimPrune (keep rate = 0.8) | 62.40±0.28 | 59% |
| Rotation + SimPrune (keep rate = 0.8) | 62.34±0.23 | 54% |

## 5.4 EXPERIMENTS ON DOWNSTREAM TASKS OF OBJECT DETECTION AND SEMANTIC SEGMENTATION

In this section, we evaluate the performance of SimPrune on two downstream tasks other than image classification: object detection and semantic segmentation. We use the MS COCO (Lin et al., 2014) dataset for the object detection task and the ADE20k (Zhou et al., 2017) dataset for the semantic segmentation task. The evaluation metrics are average precision for bounding boxes ($AP^b$) and mean IoU (mIoU), respectively. As shown in Table 6, compared to the DINO baseline, SimPrune is able to reduce 24% of the SSL computation costs with only a marginal loss in accuracy. With the same keep ratio, SimPrune delivers 1.7 higher $AP^b$ on the object detection task and 2.8 higher mIoU on the semantic segmentation task than EViT. Compared to ToMe, SimPrune offers 2.9 higher $AP^b$ and 2.3 higher mIoU. These results demonstrate the effectiveness of SimPrune across a variety of downstream tasks.

Table 6: Experiments on downstream tasks of object detection (MS COCO dataset) and semantic segmentation (ADE20K dataset). DeiT-B is used as the encoder. The keep rate is set to 0.8 for all the methods.

| Method | MS COCO | ADE20k | | |
| --- | --- | --- | --- | --- |
| | $AP^b$ | mIoU | Training FLOPs | Training Time |
| DINO | 49.8 | 34.3 | 100% | 100% |
| EViT | 48.0 | 31.2 | 76% | 89% |
| ToMe | 46.8 | 31.7 | 76% | 88% |
| SimPrune | 49.7 | 34.0 | 76% | 89% |

**Conventional Attention-based Method (EViT)**

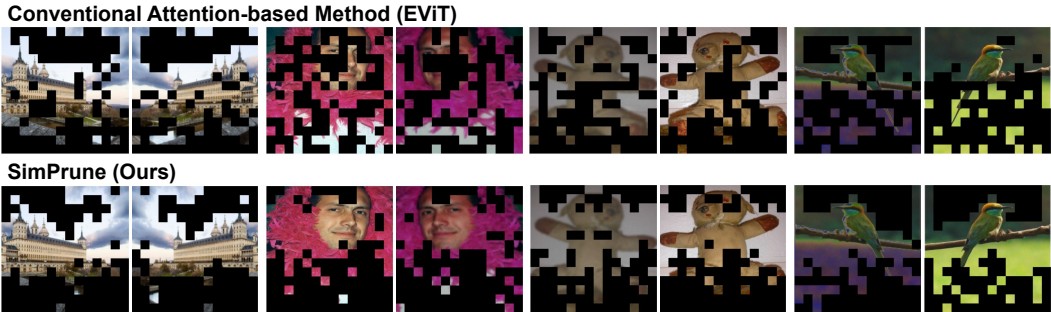

**SimPrune (Ours)**

Figure 4: Visualization of the token pruning outcomes of conventional attention-based token pruning method (EViT) and our proposed method SimPrune. The masked region represents the tokens that are pruned.

## 5.5 VISUALIZATION

We visualize the token pruning outcomes for SimPrune and the attention-based pruning method EViT in Figure 4. From the figure, we can observe that SimPrune's pruning regions are more concentrated and symmetrical compared to EViT. This is because SimPrune first matches tokens from two branches to form multiple token pairs, and then prunes at the granularity of these token pairs. As a result, tokens in a pair that have similar semantic information are either both pruned or retained, ensuring semantic consistency between the outputs of the two branches after token pruning. In contrast, if we only consider information (e.g., attention scores) from a single branch when performing token pruning, tokens with similar semantic information in different branches might be pruned differently, with one being pruned and the other retained, as discussed in 3.2. This introduces the semantic misalignment between the output representation of images in different branches. As a result, the model struggles to learn meaningful features during SSL, ultimately degrading training effectiveness and downstream performance.

## 6 CONCLUSION

In this paper, we proposed SimPrune, a novel token pruning strategy tailored specifically for ViTs in discriminative self-supervised learning. By leveraging cross-branch similarity, SimPrune efficiently prunes non-essential tokens while preserving crucial semantic consistency across different augmented views of the same image, significantly enhancing training efficiency without compromising accuracy. Additionally, we introduce a difficulty-aware pruning strategy to enhance SimPrune, which prunes token pairs at different similarity levels throughout different training stages. This strategy offers an effective and efficient way to control the training difficulty, further optimizing the model performance. Our extensive evaluation demonstrates that SimPrune can substantially reduce computation costs without compromising accuracy. Furthermore, compared to the existing popular token reduction methods such as EViT and ToMe, SimPrune achieves 3.1% and 3.2% higher accuracy, respectively, while consuming similar training costs.

ACKNOWLEDGEMENT

The authors would like to thank the anonymous reviewers for their constructive feedback and suggestions. This work is supported in part by NSF grants #2154973, #2312157, and #2334628.

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

APPENDIX

# A    EXTENDED EXPERIMENTAL RESULTS

## A.1    EXPERIMENTS ON DYNAMIC SIMPRUNE AND COMPARISON WITH OTHER TOKEN REDUCTION METHODS

In this section, we extend SimPrune to a dynamic version that can automatically prune tokens without requiring a pre-defined keep rate. Similar to other dynamic pruning approaches (Tang et al., 2022; Gao et al., 2018), we incorporate a lightweight module consisting of a downsampling layer, a linear layer, and an activation layer within each attention block. During SSL, we expand the sliding window size to include 50% of the total token pairs based on cross-branch similarity, treating these as "candidate token pairs." The module then processes the candidate tokens to determine which tokens should be pruned. As shown in Table 7, SimPrune with the dynamic pruning strategy reduces 24% computation costs without compromising accuracy compared to the DINO baseline, demonstrating that the advantages of SimPrune are preserved even when extended to a dynamic pruning framework.

We also further compare SimPrune to some other token reduction methods MCTF (Lee et al., 2024), PatchSlim (Tang et al., 2022), and SelfSlim (Zong et al., 2022). As shown in Table 7, SimPrune consistently outperforms all these methods, delivering up to 3% higher accuracy when consuming comparable computation costs.

Table 7: Experiments on Dynamic SimPrune and Comparison of other Token Reduction methods.

| Keep Rate | Method | DeiT-S | | | DeiT-B | | |
|---|---|---|---|---|---|---|---|
| | | Accuracy | Training FLOPs | Training Time | Accuracy | Training FLOPs | Training Time |
| – | DINO | 62.49 | 100% | 100% | 64.56 | 100% | 100% |
| – | SimPrune (Dynamic) | 62.52 | 78% | 87% | 64.39 | 74% | 86% |
| 0.8 | MCTF | 59.75 | 76% | 86% | 61.85 | 76% | 87% |
| | PatchSlim | 59.51 | 76% | 87% | 61.92 | 76% | 90% |
| | SelfSlim | 60.46 | 80% | 89% | 62.10 | 80% | 92% |
| | SimPrune | 62.31 | 76% | 85% | 64.21 | 76% | 89% |
| 0.7 | MCTF | 58.84 | 65% | 78% | 61.05 | 65% | 78% |
| | PatchSlim | 59.22 | 65% | 75% | 60.18 | 65% | 82% |
| | SelfSlim | 58.38 | 70% | 81% | 60.67 | 70% | 85% |
| | SimPrune | 61.90 | 65% | 76% | 63.85 | 65% | 79% |

## A.2    EXPERIMENTS ON CIFAR DATASET

In this section, we further evaluate our proposed SimPrune approach using CIFAR datasets. As shown in Table 8, SimPrune offers considerable training cost savings compared to the DINO baseline. For instance, with a keep rate of 0.8, SimPrune achieves a 24% reduction in computation FLOPs and a 14% reduction in training time without accuracy loss. Additionally, SimPrune also outperforms both EViT and ToMe, delivering 2.0% and 1.6% higher accuracy, respectively, with similar training time and FLOPs.

## A.3    EXPERIMENTS ON MOCO V3

In this section, we evaluate our proposed SimPrune method using the MoCo v3 training framework, which handles both positive and negative sample pairs. In MoCo v3, the primary objective is twofold: maximizing the similarity between two augmented views of the same input (i.e., positive sample pairs) while minimizing the similarity between views from different input samples (i.e., negative sample pairs). In this case, SimPrune performs token pruning between each positive sample pair, ensuring that essential semantic consistency is maintained across the two branches. On the other hand, since negative sample pairs inherently lack semantic connections, SimPrune focuses on the relationship between positive samples when performing token pruning. The experimental results are shown in Table 9. Compared to the MoCo baseline, SimPrune reduces training FLOPs by 24% and training

Table 8: Comparison of different methods on CIFAR dataset. DINO is used as the training framework. The encoder is DeiT-S, which is trained for 300 epochs.

| Keep Rate | Method | CIFAR-10 | | | CIFAR-100 | | |
|---|---|---|---|---|---|---|---|
| | | Accuracy | Training FLOPs | Training Time | Accuracy | Training FLOPs | Training Time |
| – | DINO | 89.54±0.13 | 100% | 100% | 62.78±0.20 | 100% | 100% |
| 0.8 | EViT | 87.75±0.26 | 76% | 84% | 61.19±0.14 | 76% | 85% |
| | ToMe | 88.41±0.14 | 76% | 86% | 62.20±0.21 | 76% | 87% |
| | SimPrune | 89.58±0.17 | 76% | 86% | 62.62±0.32 | 76% | 84% |
| 0.7 | EViT | 86.69±0.38 | 65% | 77% | 60.10±0.35 | 65% | 76% |
| | ToMe | 87.37±0.18 | 65% | 79% | 59.63±0.17 | 65% | 78% |
| | SimPrune | 89.30±0.23 | 65% | 76% | 62.32±0.24 | 65% | 77% |

Table 9: Experiments on MoCo v3 framework. DeiT-T is used as the encoders and it is trained for 100 epochs on Tiny ImageNet. The keep rate for pruning is set to 0.8.

| Method | Accuracy | Training FLOPs | Training Time |
|---|---|---|---|
| MoCo v3 | 38.46±0.17 | 100% | 100% |
| EViT | 37.65±0.24 | 76% | 85% |
| ToMe | 37.09±0.29 | 76% | 87% |
| SimPrune | 38.41±0.20 | 76% | 84% |

time by 16% without compromising accuracy. Additionally, SimPrune outperforms both EViT and ToMe, achieving 0.76% and 1.32% higher accuracy, respectively, with similar training costs. These results demonstrate the versatility and robustness of SimPrune across various self-supervised learning frameworks.

## A.4 EXPERIMENTS ON DINO V2

In this section, we evaluate our proposed SimPrune on DINOv2. DINOv2 is an enhanced discriminative SSL framework that incorporates ideas from masked image modeling (MIM) into the discriminative SSL paradigm. Specifically, DINOv2 augments the original input image into two distinct views and employs a two-part training strategy: (i) aligning the outputs of the two augmented views processed by the student and teacher branches, and (ii) simultaneously masking a subset of tokens in the view processed by the student branch and reconstructing those tokens to learn fine-grained details. This combination enables DINOv2 to leverage both global and local representation learning, enhancing its performance across various tasks. To adapt SimPrune to the DINOv2 framework, the token matching process of SimPrune first aligns the masked tokens in the view processed by the student model to their counterparts in the full view (i.e., without masking) processed by the teacher model. Subsequently, it performs token matching for the remaining tokens. This approach ensures semantic consistency and prevents misalignment by guaranteeing that the masked tokens and their corresponding counterparts are either pruned or retained together throughout the training process.

We compare our proposed SimPrune to two token reduction methods Expediting ViT (Liang et al., 2022a) and EViT. Expediting ViT reduces the number of tokens through token clustering at the early attention block and reconstructs the tokens to the original amount for further processing at the later attention block. Expediting ViT accelerates the model computation by reducing the number of tokens to be processed on the transformer blocks. It is also worth noting that both these two methods and our proposed SimPrune only embed non-parametric operators (i.e., token clustering and reconstruction, attention score calculation, and cosine similarity calculation for tokens) for token reduction. We adjust the hyper-parameter of Expediting ViT to ensure the same computation cost savings for a fair comparison. As shown in Table 10, compared to the DINOv2 baseline, SimPrune is able to reduce more than 20% of the computation with a marginal accuracy loss of 0.2%. SimPrune also outperforms both Expediting ViT and EViT, delivering 3.9% and 2.9% higher accuracy than Expediting ViT and EViT, respectively. The reason behind this is that SimPrune maintains semantic consistency across two branches during SSL. On the other hand, the token reduction methods Expediting ViT and EViT

do not consider the cross-branch similarity and semantic information when reducing the number of tokens to be processed, therefore leading to a compromised performance of the resultant model.

Table 10: Experiments on DINO v2 framework. DeiT-S is used as the encoder and it is trained on ImageNet. The keep rate is set to 0.8 for all methods.

| Method | Accuracy | Training FLOPs | Training Time |
|---|---|---|---|
| DINOv2 | 80.52 | 100% | 100% |
| Expediting ViT | 76.37 | 76% | 84% |
| EViT | 77.41 | 76% | 80% |
| SimPrune | 80.30 | 76% | 81% |

### A.5 EXPERIMENTS ON LARGER BATCH SIZE AND FULL PRECISION

In this section, for a more comprehensive evaluation, we further conduct experiments under a batch size of 1024, a learning rate of 0.002, using 8 local views, and full precision FP32. As shown in Table 11, the baseline accuracy is 77.24%, which is much better than using a batch size of 128 and half-precision. Our proposed SimPrune can still reduce the computation cost by more than 20% compared to the baseline, without compromising accuracy. Compared to the EViT and ToMe, SimPrune offers 2.0% and 2.6% higher accuracy, respectively.

Table 11: Experiments on Larger Batch Size and Full Precision. DeiT-S is used as the encoder and it is trained on ImageNet. The keep rate is set to 0.8 for all methods.

| Method | Accuracy | Training FLOPs | Training Time |
|---|---|---|---|
| DINO | 77.24 | 100% | 100% |
| EViT | 75.18 | 76% | 83% |
| ToMe | 74.60 | 76% | 85% |
| SimPrune | 77.19 | 76% | 82% |

## B OVERHEAD ANALYSIS

The computational overhead introduced by SimPrune primarily arises from the cosine similarity calculations required for token matching across the two branches. Given $N$ tokens, each represented by a $D$-dimensional vector, the cosine similarity computation between a pair of tokens can be approximated as:

$$\text{Overhead per block} = N^2 \times 2D \text{ FLOPs}$$

In our experiment setup, SimPrune is applied at three different transformer blocks, and the total overhead of SimPrune becomes:

$$\text{SimPrune Overhead} = 3 \times N^2 \times 2D \text{ FLOPs}$$

Table 12: The computational complexity of each operation in a ViT block. The input $N \times D_{ch}$ goes through three linear transformation layers with $D_{ch} \times D_{attn}$ to generate Query ($Q$), Key ($K$), and Value ($V$) matrices of size $N \times D_{attn}$. $N$ is transitive, while $D_{ch}$ is not.

| # | Module | Input Size | Operation | Layer Size | Output Size | Computation |
|---|---|---|---|---|---|---|
| ① | | $N \times D_{ch}$ | Linear Transformation | $D_{ch} \times D_{attn}$ | $N \times D_{attn}$ | $ND_{ch}D_{attn} \times 3$ |
| ② | MSA | $N \times D_{attn}$ | $Q$ Multiplying $K^T$ | - | $N \times N$ | $N^2 D_{attn}$ |
| ③ | | $N \times N$ | Multiplying $V$ | - | $N \times D_{attn}$ | $N^2 D_{attn}$ |
| ④ | | $N \times D_{attn}$ | Projection | $D_{attn} \times D_{ch}$ | $N \times D_{ch}$ | $ND_{attn}D_{ch}$ |
| ⑤ | FNN | $N \times D_{ch}$ | FC Layer | $D_{ch} \times 4D_{fc}$ | $N \times 4D_{fc}$ | $4ND_{ch}D_{fc}$ |
| ⑥ | | $N \times 4D_{fc}$ | FC Layer | $4D_{fc} \times D_{ch}$ | $N \times D_{ch}$ | $4ND_{fc}D_{ch}$ |
| | Total Computational Complexity | | | | | $4ND_{ch}D_{attn} + 2N^2 D_{attn} + 8ND_{ch}D_{fc}$ |

Table 12 shows an analysis of each operation in a Transformer block. Given an input sequence $N \times D$, where $N$ is the input sequence length or the token number and $D_{attn}$ is the embedding dimension of each token (Touvron et al., 2021). $D_c h$ is the attention layer dimension, and $D_f c$ is the dimension of the MLP layer. The total computational complexity of one block is $(4ND_{ch}D_{attn} + 2N^2 D_{attn} + 8ND_{ch}D_{fc})$. For simplicity, the computational complexity of ViT is $(12ND^2 + 2N^2 D)$ MACs.

Since there are 12 transformer blocks in a DeiT model and there are 2 branches in self-supervised learning, the computation cost for training one image in self-supervised learning can be approximated as:

$$\text{Total Cost} = 2 \times 12 \times 3 \times (12ND^2 + 2N^2 D) \text{ MACs} = 144 \times (12ND^2 + 2N^2 D) \text{ FLOPs}$$

Let us assume we are using the DeiT-Small model, then N is 196, and D is 384. So if we compare the Total overhead to the Total training cost, we can get:

$$\text{Overhead Compared to Total Cost} = \frac{3 \times N^2 \times 2D}{144 \times (12ND^2 + 2N^2 D)} \approx 0.16\%$$

As such, the overhead of our proposed method SimPrune is about 0.16% of the total computation cost, which is negligible.

## C  IMPLEMENTATION DETAILS OF HANDLING LOCAL VIEWS FOR DINO FRAMEWORK

As discussed in Section 2, the DINO framework provides an option of employing local views (smaller cropped patches) in addition to global views to enhance the model's capability to capture detailed and granular features. Specifically, while the online branch (i.e., student branch) processes both global and local views, the target branch (i.e., teacher branch) is limited to processing global views only. The online branch's output from both view types is then aligned with the global view output from the target branch. Given this setup, we first apply SimPrune between the global views of both branches. After that, we are able to obtain a set of pruned tokens for the global views in the target branch. We then perform token matching between this pruned token set and the tokens from the local views. After the matching process, we have multiple token pairs, and the local view tokens are pruned accordingly based on the similarity and keep rate.

