# OpenReview forum: "Mutual Effort for Efficiency: A Similarity-based Token Pruning for Vision Transformers in Self-Supervised Learning"
_ICLR.cc/2025/Conference — ICLR 2025 Poster_

### Official Review · Reviewer_aES2 · 2024-10-23

**Soundness:** 2
**Presentation:** 2
**Contribution:** 2
**Rating:** 6
**Confidence:** 4

**Summary:**

The authors introduce SimPrune, a novel token pruning strategy designed for ViTs in SSL. SimPrune leverages cross-branch similarity information to efficiently prune tokens, retaining essential semantic information across dual branches.

**Strengths:**

* The method is easy to understand.

**Weaknesses:**

1. Why are the results in the article so different from the official results? In the Table1, the authors use DeiT-Small as the encoder and
use ImageNet dataset to do the self-supervised training. The performance of DINO is 57.16 without pruning. But according to the [official DINO paper](https://arxiv.org/pdf/2104.14294v2), it achieves 77.0 with a ViT-S backbone. The fact that the results in this paper are so different from those in other articles makes it difficult to compare the methods in this paper to other work, so I'm curious as to what differences in setups lead to such differences.
2. Many SSL methods outperform the DINO used in this paper, such as DINOv2/MIM-Refiner/Unicom, and there are already many methods that can boost up the vision transformer without training, such as [Expediting ViT](https://openreview.net/pdf?id=9ND8fMUzOAr). So the question is, if the task is boosting up the self-supervised training process, a better framework(DINOv2/MIM-Refiner/Unicom) with a smaller backbone may be a better choice; if the task is getting a faster model, choosing a better framework then apply the prune method without finetuning, or just use a smaller network may be the better solutions. Unless the method of this paper can be shown to be effective and better than methods such as Expediting ViT on the latest self-supervised frameworks, it is difficult for me to think of where the superiority of the method of this paper lies.

**Questions:**

1. Why the performance gap exists?
2. Does the proposed method works on the latest SSL framework and achieves the competitive results?

---

> ### Author Response · Authors · 2024-11-27
> **Author Response to Reviewer aES2 (Part 1/3)**
>
> **We appreciate the valuable comments from the reviewer. We carefully address all the reviewer’s questions and revise the paper accordingly. We hope our response can help alleviate the reviewer's concern.**
>
> ---
>
> ### Q1. Why are the results in the article so different from the official results? In the Table1, the authors use DeiT-Small as the encoder and use ImageNet dataset to do the self-supervised training. The performance of DINO is 57.16 without pruning. But according to the official DINO paper, it achieves 77.0 with a ViT-S backbone. The fact that the results in this paper are so different from those in other articles makes it difficult to compare the methods in this paper to other work, so I'm curious as to what differences in setups lead to such differences.
>
> Thanks for your valuable feedback on the results.
>
> * First, let me clarify some key hyper-parameters. In our paper, the batch size in all the experiments is set to 128 (we choose 128 batch size in order not to exceed the single A100 GPU memory when training with a larger model), the learning rate is 0.001, and we adopt the half-precision FP16 (NOT full-precision FP32) during training for efficiency. We also updated Section 5.1 to show these details in the revised paper.
>
> * Second, the experimental results in Table 1 are obtained without using multi-crops (i.e., local views), as stated in the table caption and Section 3. On the other hand, the results of 77.0% accuracy in the DINO paper are obtained under the batch size of 1024, learning rate of 0.002, and using 2 global views as well as 8 local views. The **lower precision (FP16 vs. FP32)**, **smaller batch size (128 vs. 1024)**, **difference in learning rate (0.001 vs. 0.002)**, and **fewer inputs (2 global views only vs. 2 global views plus 8 local views)** could be the root cause of the difference in the resultant accuracy. DINO paper also evaluates the model performance when using smaller batch sizes. As shown in Table 9 in the DINO paper, **when adopting a batch size of 128, the reported accuracy is 57.9%, which is comparable to our result of 57.16%.**
>
> * Third, for a more comprehensive evaluation, we further **conduct experiments under a batch size of 1024, a learning rate of 0.002, using 8 local views, and full precision FP32.** The results are shown in Table R.6. As shown in the Table, the baseline accuracy is 77.24%, which is much better than using a batch size of 128 and half-precision. In this setup, our proposed SimPrune can still reduce the computation cost by more than 20% compared to the baseline, without compromising accuracy. Compared to the EViT and ToMe, SimPrune offers 2.0% and 2.6% higher accuracy, respectively. We also present the results in the revised paper in Table 11 and Appendix A.5.
>
>
> >**Table R.6: Experiments on Larger Batch Size and Full Precision. DeiT-S is used as the encoder and it
> is trained on ImageNet. The keep rate is set to 0.8 for all methods.**
> | Method                       | Accuracy    | Training FLOPs               |   Training Time            |
> |------------------------------|----------|----------------|---------------|
> | DINO                         | 77.24    | 100%            | 100%           |
> | EViT                         | 75.18    | 76%             | 83%            |
> | ToMe                         | 74.60     | 76%             | 85%            |
> | **SimPrune** | **77.19**    | **76%**             | **82%**            |
>
> * Lastly, we want to emphasize that we use **identical setups** for all the experiments in this paper to ensure a fair comparison. The purpose of this paper is not to achieve SOTA accuracy in SSL, but to present an approach to prune redundant tokens specifically in dual-branch SSL. Our proposed approach uniquely incorporates cross-branch similarity information and maintains semantic consistency, which is not addressed by single-branch token pruning methods. As such, we did not extensively tune training-related hyper-parameters (e.g., learning rate) for higher accuracy, as our primary focus is on demonstrating the effectiveness of SimPrune in comparison to other token pruning methods.

---

> ### Author Response · Authors · 2024-11-27
> **Author Response to Reviewer aES2 (Part 2/3)**
>
> ### Q2. Many SSL methods outperform the DINO used in this paper, such as DINOv2/MIM-Refiner/Unicom, and there are already many methods that can boost up the vision transformer without training, such as Expediting ViT. So the question is, if the task is boosting up the self-supervised training process, a better framework(DINOv2/MIM-Refiner/Unicom) with a smaller backbone may be a better choice; if the task is getting a faster model, choosing a better framework then apply the prune method without finetuning, or just use a smaller network may be the better solutions.
> Thanks for your constructive feedback.
>
> * First, we would like to emphasize that we focus on **accelerating dual-branch self-supervised learning (i.e., discriminative self-supervised learning)** in this paper. Dual-branch SSL (e.g., DINO) is an important and promising area in self-supervised learning research. It trains models by aligning two or more differently augmented views of the same original image, which encourages the learning of global representations. In contrast, Masked Image Modeling (MIM)-based self-supervised learning methods, such as MIM-refiner, operate by masking parts of input images and training models to reconstruct the missing regions, enabling them to capture fine-grained image details.
>
> While MIM-based methods are advantageous for tasks requiring detailed local information, dual-branch SSL methods excel in capturing global representations, making them particularly effective for some downstream tasks like image classification. For example, DINOv2 delivers **more than 2% higher accuracy** than MIM-refiner models in the downstream tasks of ImageNet classification [1].
>
> * Second, we want to emphasize that our proposed token pruning approach SimPrune **works for the backbone network of all sizes** in dual-branch SSL. We conduct experiments using three versions of DeiT model of different sizes: Tiny, Small, and Base. As shown in Table 3 on the paper. SimPrune outperforms other token pruning approaches on all three versions of DeiT.
>
> [1] Oquab, Maxime, Timothée Darcet, Théo Moutakanni, Huy V. Vo, Marc Szafraniec, Vasil Khalidov, Pierre Fernandez et al. "DINOv2: Learning Robust Visual Features without Supervision." Transactions on Machine Learning Research.

---

> ### Author Response · Authors · 2024-11-27
> **Author Response to Reviewer aES2 (Part 3/3)**
>
> ### Q3. Experiments on the latest self-supervised frameworks and Comparison to other token reduction methods.
>
> For a more comprehensive comparison, we **conduct experiments on DINOv2 frameworks.** The results are presented in Table R.7.
>
> One can observe that, our proposed **SimPrune consistently outperforms Expediting ViT and EViT, delivering 3.9% and 2.9% higher accuracies, respectively.** The reason behind this is that SimPrune maintains semantic consistency across two branches during dual-branch SSL. On the other hand, the token reduction methods Expediting ViT and EViT do not consider the cross-branch similarity and semantic information when reducing the number of tokens to be processed therefore leading to a compromised performance of the resultant model. We also provide the analysis and results in Table 10 and Appendix A.4 in the revised paper.
>
> >**Table R.7: Experiments on DINO v2 framework. DeiT-S is used as the encoder and it is trained on ImageNet. The keep rate is set to 0.8 for all methods.**
> | Method                       | Accuracy | Training FLOPs | Training Time |
> |------------------------------|----------|----------------|---------------|
> | DINOv2                       | 80.52    | 100%            | 100%           |
> | Expediting ViT                  | 76.37    | 76%             | 83%            |
> | EViT                         | 77.41    | 76%             | 80%            |
> | **SimPrune** | **80.3**     | **76%**             | **81%**            |
>
>
> **Details on DINOv2 and Expediting ViT:**
>
> DINOv2 is an enhanced dual-branch SSL framework (i.e., discriminative SSL) that **incorporates ideas from masked image modeling (MIM) into the dual-branch SSL paradigm**. Specifically, DINOv2 augments the original input image into two distinct views and employs a two-part training strategy: (i) aligning the outputs of the two augmented views processed by the student and teacher branches, and (ii) simultaneously masking a subset of tokens in the view processed by the student branch and reconstructing those tokens to learn fine-grained details. This combination enables DINOv2 to leverage both global and local representation learning, enhancing its performance across various tasks.
>
> To adapt SimPrune to the DINOv2 framework, the token matching process of SimPrune first aligns the masked tokens in the view processed by the student model to their counterparts in the full view (i.e., without masking) processed by the teacher model. Subsequently, it performs token matching for the remaining tokens. This approach ensures semantic consistency and prevents misalignment by guaranteeing that the masked tokens and their corresponding counterparts are either pruned or retained together throughout the training process.
>
>
> We compare our proposed SimPrune to two token reduction methods Expediting ViT and EViT. Expediting ViT reduces the number of tokens through token clustering at the early attention block and reconstructs the tokens to the original amount for further processing at the later attention block. Expediting ViT accelerates the model computation by reducing the number of tokens to be processed on the transformer blocks. EViT is a recognized attention-based token pruning method that prunes tokens with lower attention scores. It is also worth noting that **both these two methods and our proposed SimPrune only embed non-parametric operators** (i.e., token clustering and reconstruction, attention score calculation, and cosine similarity calculation for tokens) for token reduction.

---

> > ### Author Response · Authors · 2024-12-02
> > **Author Response on more experimental results (semantic segmentation) using the latest SSL framework**
> >
> > ### Q3. Experiments on downstream tasks of semantic segmentation using the latest self-supervised frameworks and Comparison to other token reduction methods. **(cont.)**
> >
> > For a more comprehensive comparison, **in addition to image classification on ImageNet**, we also **conduct experiments on downstream tasks of semantic segmentation using DINOv2 frameworks.** The results are presented in Table R.8.
> >
> > As shown in the table, our proposed **SimPrune consistently outperforms Expediting ViT and EViT, providing 2.1 and 2.3 higher mIoU, respectively.** These results show the superior performance of SimPrune across a variety of downstream tasks.
> >
> > >**Table R.8: Experiments on downstream tasks of semantic segmentation. DINO v2 is the SSL framework. DeiT-S is used as the encoder and it is trained on ImageNet. The keep rate is set to 0.8 for all methods.**
> > | Method  | CityScapes |  ADE20k    |                | |
> > |-------------|--------|------|----------------|----------------|
> > |       | mIoU   | mIoU | Training FLOPs | Training Time |
> > | DINOv2      | 65.1   | 43.3 | 100%            | 100%           |
> > | Expedit ViT | 62.8   | 41.2 | 76%             | 83%            |
> > | EViT        | 63.0     | 40.5 | 76%             | 80%            |
> > | **SimPrune**    | **65.0**     | **43.1** | **76%**             | **81%**            |

---

> ### Author Response · Authors · 2024-12-02
> **Author Response to Reviewer aES2**
>
> Dear Reviewer aES2,
>
> Thanks for your valuable time and reviewing efforts! We appreciate your constructive comments.
>
> We have provided suggested results in the authors' response, such as the experiments on other hyper-parameters for higher accuracy, experiments on the latest framework, comparisons with other SOTA works on the latest framework, and experiments on more downstream tasks other than ImageNet classification. We have also provided detailed clarification and explanation of your concerns about the performance gap.
>
> We hope our responses have answered your questions. It would be our great pleasure if you would consider updating your review or score.
>
> Best,
>
> Authors

---

### Official Review · Reviewer_JVJ6 · 2024-10-30

**Soundness:** 3
**Presentation:** 3
**Contribution:** 3
**Rating:** 6
**Confidence:** 3

**Summary:**

This paper first analyzes the effectiveness of conventional single-branch token pruning frameworks on SSL and reveals these methods fail to efficiently prune tokens for SSL approaches. To alleviate this issue, this paper proposes to guide token pruning based on cross-branch similarity. Besides, a difficulty-aware pruning strategy is introduced to control the difficulty of the training process. Experiments are conducted to verify the effectiveness of the proposed method.

**Strengths:**

1. The motivation is clear. The finding that existing token pruning strategies fail to enhance SSL efficiencies is interesting. It is reasonable to prune the pair tokens from two branches based on the cross-branch similarities in the SSL paradigm.

2. The conducted experiments and visualizations are extensive and well-organized.

3. The paper is well-written and easy to understand.

**Weaknesses:**

1. Some important references about token pruning are missing:
[1] Not all images are worth 16x16 words: Dynamic transformers for efficient image recognition, NeurIPS 2021.
[2] Patch slimming for efficient vision transformers, CVPR 2022.
[3] Self-slimmed vision transformer, ECCV 2022.

2. The downstream tasks, such as object detection and semantic segmentation, are widely adopted to verify the effectiveness of SSL methods (e.g., MAE and ViTDet). Could you present some finetuning results on downstream tasks?

3. Given that some layers of the ViT only observe a subset of tokens during SSL training due to token pruning, a potential discrepancy arises: downstream tasks typically utilize all tokens. Does this inconsistency decrease the downstream performance of SSL-trained models?

4. What's the sliding window size in your experiments?

5. This paper utilizes a dynamic pruning strategy. The visualizations shown in Figure 4 are static. Could you provide additional visualizations, statistics, or other observations illustrating how the pruning patterns change over time?

**Questions:**

Please see the Weakness section.

---

> ### Author Response · Authors · 2024-11-27
> **Author Response to Reviewer JVJ6 (Part 1/2)**
>
> **We would like to thank the reviewer for the positive feedback and valuable questions. We appreciate the reviewer's acknowledgment that our motivation is clear, our proposed work is reasonable, and the experimental results are extensive and well-organized. We carefully address all the reviewer’s questions and revise the paper accordingly. We hope our response can help clarify the reviewer's questions.**
>
> ---
>
> ### Q1. Some important references about token pruning are missing.
>
> Thanks for pointing out these important prior works. We have referenced these works in our paper in Section 5.3 in the revised paper. **We also compare our method with “Patch slimming for efficient vision transformers” (PatchSlim) and “Self-slimmed vision transformer” (SelfSlim).** The results are shown in Table R.4.
>
> Our proposed SimPrune **achieves 2.9% and 2.7% higher accuracy compared to PatchSlim and SelfSlim**, while consuming similar computation costs under the same keep rate. The superior performance of our proposed SimPrune stems from the maintenance of semantic consistency across two branches during dual-branch SSL. On the other hand, the token pruning methods (PatchSlim and SelfSlim) do not consider the cross-branch similarity and semantic information when performing token pruning therefore leading to a compromised performance of the resultant model. We also provide the results in Table 5 and Section 5.3 in the revised paper.
>
> “Not all images are worth 16x16 words” is not a method that targets reducing computation cost in training but reducing computation cost in inference. In fact, it increases the training cost as it has to train multiple vision transformers. Since our paper focuses on dual-branch self-supervised training, we do not compare our approach to this work.
>
>
> >**Table R.4: Comparison of two other Token Reduction methods.**
> | Keep Rate | Method                       | DeiT-S   |                |               | DeiT-B   |                |               |
> |-----------|------------------|----------|----------------|---------------|----------|----------------|---------------|
> |           |                              | Accuracy | Training FLOPs | Training Time | Accuracy | Training FLOPs | Training Time |
> |           | DINO                         | 62.49    | 100%            | 100%           | 64.56    | 100%            | 100%           |
> | 0.8       | PatchSlim                    | 59.51    | 76%             | 87%            | 61.92    | 76%             | 90%            |
> |           | SelfSlim                     | 60.46    | 80%             | 89%            | 62.10     | 80%             | 92%            |
> |           | **SimPrune** | **62.31**    | **76%**             | **85%**            | **64.21**    | **76%**             | **89%**            |
> | 0.7       | PatchSlim                    | 59.22    | 65%             | 75%            | 60.18    | 65%             | 82%            |
> |           | SelfSlim                     | 58.38    | 70%             | 81%            | 60.67    | 70%             | 85%            |
> |           | **SimPrune** | **61.90**     | **65%**             | **76%**            | **63.85**    | **65%**             | **79%**            |
>
>
>
>
> ---
>
> ### Q2. Experiments on downstream tasks of object detection and semantic segmentation.
>
> Thanks for your valuable feedback on conducting experiments on more downstream tasks. **We have conducted the suggested experiments of object detection and semantic segmentation.** The results are shown in Table R.5. We use MS COCO dataset for the object detection task and ADE20k dataset for the semantic segmentation task. The evaluation metrics are average precision for bounding boxes (APb) and mean IoU (mIoU), respectively.
>
> Compared to the DINO baseline, SimPrune is able to reduce computation costs with only a marginal loss in accuracy. Compared to EViT, SimPrune delivers 1.7 higher APb on object detection task and 2.8 higher mIoU on semantic segmentation task. Compared to ToMe, SimPrune offers 2.9 higher APb and 2.3 higher mIoU. These results demonstrate the effectiveness of SimPrune across a variety of downstream tasks. We also present the results in Table 7 and Appendix A.1 in the revised paper.
>
>
> >**Table R.5: Experiments on downstream tasks of object detection and semantic segmentation. DeiT-B is used as the encoder. The keep rate is set to 0.8 for all methods.**
> | Method                       | COCO | ADE20k |                |               |
> |------------|------|--------|---------|---------------|
> |                              | APb  | mIoU   | Training FLOPs | Training Time |
> | DINO                         | 49.8 | 34.3   | 100%            | 100%           |
> | EViT                         | 48.0   | 31.2   | 76%             | 89%            |
> | ToMe                         | 46.8 | 31.7   | 76%             | 88%            |
> | **SimPrune** | **49.7** | **34.0**     | **76%**             | **89%**            |

---

> ### Author Response · Authors · 2024-11-27
> **Author Response to Reviewer JVJ6 (Part 2/2)**
>
> ### Q3. Given that some layers of the ViT only observe a subset of tokens during SSL training due to token pruning, a potential discrepancy arises: downstream tasks typically utilize all tokens. Does this inconsistency decrease the downstream performance of SSL-trained models?
>
> Thanks for your valuable question. We want to point out that this discrepancy would **NOT** affect the model performance in the downstream task.
>
> The underlying philosophy of token pruning is to prune or merge tokens that contain highly similar information. In other words, the tokens that represent highly similar features are redundant so that they can be pruned or merged to reduce computation costs and redundancy. By removing redundant tokens, the model is encouraged to concentrate on the most salient features during training, which can enhance its ability to generalize. As such, pruning redundant tokens during SSL would not affect the downstream task performance.
>
> Our experimental results (see Tables 3~11 in the paper) also demonstrate that our proposed token pruning approach SimPrune would not affect the performance in downstream tasks including image classification, object detection, and image segmentation.
>
>
> ---
>
>
>
> ### Q4. What's the sliding window size in your experiments?
>
> Thank you for your important question. The sliding window size is determined by **the user-defined keep rate, which is typically set based on the desired level of computation savings for a specific application.** For example, if the keep rate is set to 0.8 and the current attention block contains 100 tokens, the sliding window size will be 80 tokens. This means that only the tokens within the sliding window will be retained for further processing.
>
> ---
>
> ### Q5. Additional visualizations to illustrate how the pruning patterns change over time?
>
> Thanks for your valuable suggestions. We provide the additional visualizations (Figure 5) in Appendix D in the revised paper. As observed, the pruned regions in our proposed SimPrune **are more symmetric and concentrated across the two branches** compared to the pruned regions of single-branch token pruning approaches throughout the entire training process. This suggests our proposed SimPrune is able to maintain the semantic consistency across two branches.

---

> > ### Comment · Reviewer_JVJ6 · 2024-12-02
> >
> > Thanks for your response. I keep my initial score.

---

> > > ### Author Response · Authors · 2024-12-02
> > > **Author Response to Reviewer JVJ6**
> > >
> > > Thank you for your positive feedback and constructive comments (e.g., providing more results on downstream tasks of object detection and semantic segmentation), which make our paper stronger. We have addressed all your comments in our revision. Thank you again for your valuable time.
> > >
> > > Best,
> > >
> > > Author

---

### Official Review · Reviewer_8Qhp · 2024-10-31

**Soundness:** 2
**Presentation:** 3
**Contribution:** 2
**Rating:** 6
**Confidence:** 4

**Summary:**

This paper first conduct a preliminary study to analyze the effectiveness of conventional single-branch token pruning on dual-branch self-supervised learning (SSL) for vision transformers. Then, the authors propose SimPrune, which utilizes cross-branch similarity to guide token pruning and introduce a difficulty-aware pruning strategy to further enhance the approach. Experimental results demonstrate the effectiveness of the proposed SimPrune.

**Strengths:**

1. This paper is well-written. The insights, methodology and experimental results are introduced very clearly.
2. The preliminary study of the effectiveness of conventional single-branch token pruning on dual-branch SSL can provide some valuable insights for researchers in this field.
3. The proposed approach SimPrune seems to be somewhat novel.
4. The experimental results demonstrate the effectiveness of SimPrune on image classification tasks.

**Weaknesses:**

1. The related work section is incomplete. There are other works, such as BeiT [1], MAE [2], and SimMIM [3], which also claim to be performing SSL for vision transformers. Although these are not dual-branch methods, the relationship between these methods and the proposed SimPrune should be clarified. For instance, it should be discussed whether dual-branch SSL approaches are better than MIM-based ones (e.g., on accuracy, speed or training costs), thereby highlighting the significance of this work's contribution.
2. The experiment section is incomplete. This paper only evaluates the effectiveness of SimPrune on image classification tasks. However, in classification tasks, local features may not be critical, which is quite different from other dense tasks like object detection and image segmentation. The effectiveness of this token-pruning based approach should be further evaluated on those tasks (e.g., COCO, ADE20K, etc) to demonstrate the significance.
3. The significance of this work is not very clear. It seems that saving such training costs of performing dual-branch SSL for these small models is not critical in this field, but the effectiveness on large models is not verified yet.

I will pay more attention to the first two concerns since the third one is not actionable in a short time.

[1] Bao, Hangbo, Li Dong, Songhao Piao, and Furu Wei. "BEiT: BERT Pre-Training of Image Transformers,” In ICLR, 2022. \
[2] He, Kaiming, Xinlei Chen, Saining Xie, Yanghao Li, Piotr Dollár, and Ross Girshick. "Masked autoencoders are scalable vision learners." In CVPR, 2022. \
[3] Xie, Zhenda, Zheng Zhang, Yue Cao, Yutong Lin, Jianmin Bao, Zhuliang Yao, Qi Dai, and Han Hu. "Simmim: A simple framework for masked image modeling." In CVPR, 2022.

**Questions:**

1. The proposed similarity-based token pair pruning approach seems to be a little complicated due to many-to-one issue. Have you tried some other methods like bipartite matching?
2. Is “24% savings in training costs” significant enough? It seems that the other pruning based methods mentioned in this paper can save about 30%-40% costs.

---

> ### Author Response · Authors · 2024-11-27
> **Author Response to Reviewer 8Qhp (Part 1/2)**
>
> **We would like to thank the reviewer for the constructive feedback. We appreciate the reviewer's acknowledgment that our preliminary study is insightful and our proposed work is novel. We have added more results as suggested by the reviewer, including the experimental results on the object detection and semantic segmentation downstream tasks. We hope our response can help clarify the reviewer's questions.**
>
> ---
>
>
> ### Q1. Incomplete related works on MIM-based SSL.
>
> Thanks for your valuable suggestions on the missing related works. We have **added a brief introduction of MIM-based SSL and the advantages of MIM-based and dual-branch (i.e., Discriminative) SSL in both introduction and related works sections**.
>
> Specifically, MIM-based methods train models by reconstructing masked parts of input images, enabling them to capture fine-grained details that are beneficial for tasks like object detection and segmentation. In contrast, dual-branch SSL methods learn representations by aligning differently augmented views of the same image, focusing on global features rather than local reconstruction. This makes them particularly effective for classification tasks. For example, the DINO framework delivers about **3% higher accuracy** on the downstream task of ImageNet classification compared to MIM-based methods MAE [1]. In this paper, we focus on dual-branch SSL, an important and promising area in self-supervised learning research.
>
> [1] He, Kaiming, Xinlei Chen, Saining Xie, Yanghao Li, Piotr Dollár, and Ross Girshick. "Masked autoencoders are scalable vision learners." In CVPR, 2022.
>
>
> * Moreover, **for a more comprehensive comparison, we also conduct experiments on the DINOv2 framework**, an enhanced dual-branch SSL framework that **incorporates ideas from masked image modeling (MIM) into the dual-branch SSL paradigm**. The experimental results in shown in Table R.2. One can observe that our proposed SimPrune consistently outperforms Expediting ViT and EViT, delivering 3.9% and 2.9% higher accuracies, respectively. More details of DINOv2 and the compared method Expediting ViT are shown below in this question.
>
> The reason behind the superior performance of SimPrune is that it maintains semantic consistency across two branches during dual-branch SSL. On the other hand, the token reduction methods Expediting ViT and EViT do not consider the cross-branch similarity and semantic information when reducing the number of tokens to be processed therefore leading to a compromised performance of the resultant model. We also provide the analysis and results in Table 10 and Appendix A.4 in the revised paper.
>
> >**Table R.2: Experiments on DINO v2 framework. DeiT-S is used as the encoder and it is trained on ImageNet. The keep rate is set to 0.8 for all methods.**
> | Method                       | Accuracy | Training FLOPs | Training Time |
> |------------------------------|----------|----------------|---------------|
> | DINOv2                       | 80.52    | 100%            | 100%           |
> | Expediting ViT                  | 76.37    | 76%             | 83%            |
> | EViT                         | 77.41    | 76%             | 80%            |
> | **SimPrune** | **80.3**     | **76%**             | **81%**            |
>
>
> **Details of DINOv2 and compared methods:**
>
> Specifically, DINOv2 augments the original input image into two distinct views and employs a two-part training strategy: (i) aligning the outputs of the two augmented views processed by the student and teacher branches, and (ii) simultaneously masking a subset of tokens in the view processed by the student branch and reconstructing those tokens to learn fine-grained details. This combination enables DINOv2 to **leverage both global and local representation learning,** enhancing its performance across various tasks.
>
> To adapt SimPrune to the DINOv2 framework, the token matching process of SimPrune first aligns the masked tokens in the view processed by the student model to their counterparts in the full view (i.e., without masking) processed by the teacher model. Subsequently, it performs token matching for the remaining tokens. This approach ensures semantic consistency and prevents misalignment by guaranteeing that the masked tokens and their corresponding counterparts are either pruned or retained together throughout the training process.
>
> We compare our proposed SimPrune to two token reduction methods Expediting ViT and EViT. Expediting ViT reduces the number of tokens through **token clustering at the early attention block and reconstructs the tokens to the original amount for further processing at the later attention block.** Expediting ViT accelerates the model computation by reducing the number of tokens to be processed on the transformer blocks. EViT is a recognized attention-based token pruning method that prunes tokens with lower attention scores.

---

> ### Author Response · Authors · 2024-11-27
> **Author Response to Reviewer 8Qhp (Part 2/2)**
>
> ### Q2. Experiments on downstream tasks of object detection and semantic segmentation.
>
> Thanks for your valuable feedback on conducting more comprehensive experiments. **We have conducted the suggested experiments of object detection and image segmentation.** The results are shown in Table R.3. We use MS COCO dataset for the object detection task and ADE20k dataset for the semantic segmentation task. The evaluation metrics are average precision for bounding boxes (APb) and mean IoU (mIoU), respectively.
>
> Compared to the DINO baseline, SimPrune is able to reduce computation costs with only a marginal loss in accuracy. Compared to EViT, SimPrune delivers 1.7 higher AP^b on object detection task and 2.8 higher mIoU on semantic segmentation task. Compared to ToMe, SimPrune offers 2.9 higher APb and 2.3 higher mIoU. These results demonstrate the effectiveness of SimPrune across a variety of downstream tasks. We also present the results in Table 7 and Appendix A.1 in the revised paper.
>
> >**Table R.3: Experiments on downstream tasks of object detection and semantic segmentation. DeiT-B is used as the encoder. The keep rate is set to 0.8 for all methods.**
> | Method                       | COCO | ADE20k |                |               |
> |------------------------------|------|--------|----------------|---------------|
> |                              | APb  | mIoU   | Training FLOPs | Training Time |
> | DINO                         | 49.8 | 34.3   | 100%            | 100%           |
> | EViT                         | 48.0   | 31.2   | 76%             | 89%            |
> | ToMe                         | 46.8 | 31.7   | 76%             | 88%            |
> | **SimPrune** | **49.7** | **34.0**     | **76%**             | **89%**            |
>
>
> ---
>
>
> ### Q3. Experiments on larger model.
>
> Thank you for your valuable feedback. In our experiments, we included the base version of the DeiT model. DeiT-Base, with approximately 86 million parameters, requires about 5 days of training on the ImageNet dataset using a 4×A100 GPU setup. Our proposed SimPrune reduces the training time by over one day without compromising accuracy. Unfortunately, training larger models, such as DeiT-Large, exceeds the computational resources currently available to us. We sincerely appreciate your understanding and will make every effort to provide additional results on larger models in the final version of the paper if it is accepted.
>
>
> ---
>
> ### Q4. The proposed similarity-based token pair pruning approach seems to be a little complicated due to many-to-one issues. Have you tried some other methods like bipartite matching?
>
> Thanks for your insightful questions.
>
> * First, we want to emphasize that the many-to-one issue in our similarity-based token pair pruning approach is acceptable and not severe. In practice, we observe that the number of tokens pruned in the two branches typically differs by **less than 10%**, which has a negligible impact on performance.
>
> * Second, while we agree that bipartite matching is an appealing alternative, we have experimented with such methods, including the Hungarian algorithm and greedy matching. Unfortunately, although these algorithms can ensure one-to-one matching, their overhead is **more than 100× higher** than our adopted token matching method. In fact, the time required for token matching with bipartite matching is **even longer than the time it would take to process all tokens directly with the model**. Therefore, ensuring exact one-to-one matching is not computationally infeasible.
>
>
> ---
>
> ### Q5. Is “24% savings in training costs” significant enough? It seems that the other pruning based methods mentioned in this paper can save about 30%-40% costs.
>
> Thanks for the important question. We want first to mention that in our experiments (Table 3~11 in the paper), SimPrune, which maintains semantic consistency across two branches during SSL, **consistently outperforms token pruning approaches designed for supervised learning under various prune ratios.**
>
> Second, SimPrune achieves a 24% cost savings (with a keep rate of 0.8) without any accuracy loss, as demonstrated in Table 3. Furthermore, for the same 24% cost savings, SimPrune delivers **more than 2% higher accuracy** compared to EViT and ToMe.
>
> When the keep rate is reduced from 0.8 to 0.7 (i.e., increasing the prune ratio), SimPrune achieves 35% computation savings with a 0.5% accuracy drop compared to the DINO baseline. In contrast, EViT and ToMe experience **up to 4% accuracy loss** under the same cost savings.

---

> ### Comment · Reviewer_8Qhp · 2024-12-02
> **Response to Author**
>
> Thanks for the detailed reply. The author has made a good rebuttal and most of my concerns have been addressed. I will raise my rating to 6.

---

> > ### Author Response · Authors · 2024-12-02
> > **Author Response to Reviewer 8Qhp**
> >
> > Thank you for raising the score and your time spent reviewing our paper. This is a great affirmation of our work. Your comments are very constructive (e.g., providing more results on downstream tasks like object detection and semantic segmentation), which makes our paper stronger. We have addressed your comments in our revision. Thank you again for your valuable time.
> >
> > Best,
> >
> > Author

---

### Official Review · Reviewer_1ySw · 2024-11-04

**Soundness:** 2
**Presentation:** 3
**Contribution:** 3
**Rating:** 6
**Confidence:** 2

**Summary:**

This paper proposes Similarity-Based Pruning Method (SimPrune), a token pruning technique specifically designed for Vision Transformers in self-supervised learning (SSL). Unlike traditional pruning methods that rely on single-branch self-attention mechanisms, SimPrune uses cross-branch similarity to select tokens for pruning, preserving crucial cross-branch semantic consistency in SSL, thereby avoiding loss of important information due to improper pruning.

Experiments show that SimPrune can reduce training costs by approximately 24% while maintaining accuracy.

**Strengths:**

1. SimPrune is compatible and tailored for the dual-branch Siamese architecture used in SSL, where each branch processes different augmented versions of the same image. SimPrune ensures token consistency across branches, preventing unnecessary information loss, which is challenging to achieve with single-branch pruning. This makes it especially suitable for SSL.

2. SimPrune introduces a “difficulty adjustment during training” pruning strategy. In the early stages, it retains token pairs with high similarity, allowing the model to learn simpler patterns. As training progresses, it prunes increasingly similar tokens, enhancing the learning challenge. This design, inspired by the concept of “curriculum learning”, helps improve the model’s understanding of complex features.

**Weaknesses:**

I believe this paper has some interesting ideas on self-supervised token pruning.
As far as I know, the current SoTA is: Multi-criteria Token Fusion with One-step-ahead Attention for Efficient Vision Transformers, CVPR2024.

However, the performance is still inferior to the supervised token pruning.
And the proposed method requires many additional computation steps.

1. In Section 3.2, “Applying Existing Token Pruning Approach to SSL,” the authors note that SSL accuracy drops significantly when traditional self-attention pruning methods are applied. The experiments reveal that SSL is highly sensitive to token pruning; even slight over-pruning leads to substantial accuracy loss compared to supervised learning (pp. 5-6).

This sensitivity arises because SSL relies more heavily on feature consistency than supervised learning, making it vulnerable to inappropriate pruning. To avoid compromising model performance, SimPrune requires precise tuning of hyperparameters like pruning ratios and retention rates.


2. In Section 4.2, “Difficulty-Aware Pruning Strategy,” the authors propose a pruning approach that gradually increases in difficulty throughout training. Initially, token pairs with high similarity are retained, but as training progresses, more similar tokens are pruned (p. 6).

SimPrune’s design, which progressively raises training difficulty, challenges the model to handle increasingly complex features in later stages. If the model struggles to adapt to this increased difficulty, it may experience fluctuations or even declines in accuracy.


3. In Section 4.1, “Leveraging Cross-Branch Similarity for Token Pruning,” the authors explain that SimPrune involves calculating cross-branch similarities by matching tokens using cosine similarity to establish token pairs across branches (pp. 5-6).

SimPrune’s requirement for cross-branch similarity calculations introduces additional cosine similarity computations. While this overhead is minor relative to the overall computational cost, it can increase total runtime in resource-constrained environments.


4. In Section 4, “SimPrune Design,” the authors highlight that SimPrune includes key parameters, such as the “token keep rate,” which significantly affect final accuracy (p. 6).

SimPrune’s performance is highly sensitive to parameters like token retention rates and pruning stages, requiring careful adjustment based on the dataset and training setup. This need for customization adds complexity to implementation.


5. In Section 3.2, the authors note that traditional pruning methods do not ensure semantic consistency across branches, potentially resulting in a loss of cross-branch semantic information (p. 5).

While SimPrune seeks to maintain semantic consistency through cross-branch pruning, low precision in token matching may still cause semantic inconsistencies between branches, which can negatively impact model performance.


6. In Section 5.3, “Compatibility of SimPrune with Other Efficient SSL Methods,” the authors observe that SimPrune demands high computational resources in the early stages due to extensive token matching and pruning operations needed to maintain semantic consistency (pp. 8-9).

During initial training, SimPrune’s intensive token matching and pruning calculations lead to high resource demands. While these requirements lessen in later stages, the early computational load may pose challenges for resource-limited devices or environments.

**Questions:**

Please see weakness comments and I would like to see the author's response if I have interpreted these correctly.

---

> ### Author Response · Authors · 2024-11-27
> **Author Response to Reviewer 1ySw (Part 1/3)**
>
> **We would like to thank the reviewer for the valuable feedback. We carefully address the questions raised by the reviewer. We have also added more results as suggested by the reviewer, including the comparison with a SOTA token pruning work, and the experimental results of extending our proposed work to a dynamic pruning strategy. We hope our response can help clarify the reviewer's questions.**
>
> ---
>
> ### Q1. I believe this paper has some interesting ideas on self-supervised token pruning. As far as I know, the current SoTA is: Multi-criteria Token Fusion with One-step-ahead Attention for Efficient Vision Transformers, CVPR2024.
>
> Thanks for pointing out the SOTA token pruning/fusion work MCTF. MCTF fuses the token based on the token similarity, informativeness (attention score), and size (the number of fused tokens).
>
> **We compare our proposed SimPrune to MCTF**. The results are shown in Table R.1. One can observe that our proposed SimPrune achieves 2.7% higher accuracy with similar computation costs under the same prune ratio. The superior performance of SimPrune stems from its consideration of cross-branch semantic consistency and similarity information. We also add the results to Table 5 and Section 5.3 in the revised paper.
>
> >**Table R.1: Experiments on Dynamic SimPrune and Comparison of another Token Reduction method MCTF.**
> | Keep Rate | Method                       | DeiT-S   |                |               | DeiT-B   |                |               |
> |-----------|------------------------------|----------|----------------|---------------|----------|----------------|---------------|
> |           |                              | Accuracy | Training FLOPs | Training Time | Accuracy | Training FLOPs | Training Time |
> | --        | DINO                         | 62.49    | 100%            | 100%           | 64.56    | 100%            | 100           |
> | --        | **SimPrune (Dynamic)**           | **62.52**    | **78%**             | **87%**            | **64.39**    | **74%**             | **86%**            |
> | 0.8       | MCTF                         | 59.75    | 76%             | 86%            | 61.85    | 76%             | 87%            |
> |           | **SimPrune** | **62.31**    | **76%**             | **85%**            | **64.21**    | **76%**             | **89%**            |
> | 0.7       | MCTF                         | 58.84    | 65%             | 78%            | 61.05    | 65%             | 78%            |
> |           | **SimPrune** | **61.90**     | **65%**             | **76%**            | **63.85**    | **65%**             | **79%**            |

---

> ### Author Response · Authors · 2024-11-27
> **Author Response to Reviewer 1ySw (Part 2/3)**
>
> ### Q2. To avoid compromising model performance, SimPrune requires precise tuning of hyperparameters like pruning ratios and retention rates.
>
> Thanks for your valuable feedback.
>
> * **First, it is important to emphasize that we do NOT introduce additional hyper-parameters** in our approach. The prune ratio, a widely adopted user-defined hyper-parameter in token pruning methods such as EViT and ToMe [1][2], is typically **configured based on the desired level of computation savings.** This parameter allows for flexibility across different applications: for accuracy-sensitive scenarios, the prune ratio can be set conservatively to minimize accuracy loss, while in less accuracy-critical applications, a higher prune ratio can be applied to achieve greater computation savings.
>
> To showcase the superiority of our proposed SimPrune **across varying levels of computation saving**, we adopt the experimental setup used in recognized prior token pruning methods, such as EViT and ToMe, which evaluate accuracy and computation cost under different prune ratios (reversely indicated by the keep rate). Our extensive experimental results demonstrate that SimPrune, designed to preserve semantic consistency when pruning tokens in dual-branch SSL, consistently outperforms other approaches, achieving significantly higher accuracy—up to more than 3%—across all tested prune ratios.
>
>
> * **Second, the dynamic pruning strategy that can automatically prune tokens without a pre-defined prune rate is complementary to our proposed SimPrune** and our approach can be **easily extended to a dynamic version**. Specifically, we follow most other dynamic pruning approaches [3][4] and incorporate a lightweight module consisting of a downsampling layer, a linear layer, and an activation layer within each attention block. During SSL, we expand the sliding window size to include 50\% of the total token pairs based on cross-branch similarity, treating these as “candidate token pairs.” The module then processes the candidate tokens to determine which tokens should be pruned.
>
> **We present the experimental results of dynamic SimPrune in Table R.1.** As shown in the table, SimPrune with the dynamic pruning strategy reduces 24% computation costs without compromising accuracy compared to the DINO baseline. It demonstrates that the advantages of our proposed SimPrune are maintained when it is extended to a dynamic pruning strategy. We also present the results in Table 5 and Section 5.3 in the revised paper.
>
>
> [1] Liang, Youwei, G. E. Chongjian, Zhan Tong, Yibing Song, Jue Wang, and Pengtao Xie. "EViT: Expediting Vision Transformers via Token Reorganizations." In International Conference on Learning Representations.
>
> [2] Bolya, Daniel, Cheng-Yang Fu, Xiaoliang Dai, Peizhao Zhang, Christoph Feichtenhofer, and Judy Hoffman. "Token Merging: Your ViT But Faster." In The Eleventh International Conference on Learning Representations.
>
> [3] Tang, Yehui, Kai Han, Yunhe Wang, Chang Xu, Jianyuan Guo, Chao Xu, and Dacheng Tao. "Patch slimming for efficient vision transformers." In Proceedings of the IEEE/CVF Conference on Computer Vision and Pattern Recognition, pp. 12165-12174. 2022.
>
> [4] Gao, Xitong, Yiren Zhao, Łukasz Dudziak, Robert Mullins, and Cheng-zhong Xu. "Dynamic channel pruning: Feature boosting and suppression." arXiv preprint arXiv:1810.05331 (2018).
>
>
> ---
>
>
> ### Q3. SimPrune’s design, which progressively raises training difficulty, challenges the model to handle increasingly complex features in later stages. If the model struggles to adapt to this increased difficulty, it may experience fluctuations or even declines in accuracy.
>
> Thanks for your insightful question. We want to point out that according to the curriculum learning theory [5][6], if the training takes effect, the model will gradually converge and adapt to more challenging tasks, and it could even benefit from the increased training difficulty in the late training stages. As such, if the training setting like training epochs is set appropriately, SimPrune’s strategy that gradually increases the training difficulty would **NOT** hurt the accuracy. Instead, it can help the model accuracy, as shown in Section 4.2 and Table 2 in the paper.
>
> [5] Zheng, Wenzhao, Zhaodong Chen, Jiwen Lu, and Jie Zhou. "Hardness-aware deep metric learning." In Proceedings of the IEEE/CVF conference on computer vision and pattern recognition, pp. 72-81. 2019.
>
> [6] Ma, Jiachen, Yong Liu, Meng Liu, and Meng Han. "Curriculum contrastive learning for fake news detection." In Proceedings of the 31st ACM International Conference on Information & Knowledge Management, pp. 4309-4313. 2022.

---

> ### Author Response · Authors · 2024-11-27
> **Author Response to Reviewer 1ySw (Part 3/3)**
>
> ### Q4. SimPrune’s requirement for cross-branch similarity calculations introduces additional cosine similarity computations. While this overhead is minor relative to the overall computational cost, it can increase total runtime in resource-constrained environments.
>
> Thank you for your valuable feedback regarding the overhead. Based on our analysis in Appendix B and corresponding experimental results, the additional overhead introduced is **less than 1%** of the total cost, encompassing both computation and memory access. We use cosine similarity as the metric for token matching, which is computationally lightweight. Moreover, this overhead has already been accounted for in the computation cost and time measurements reported in our experiments.
>
>
> ---
>
>
>
> ### Q5. SimPrune’s performance is highly sensitive to parameters like token retention rates and pruning stages, requiring careful adjustment based on the dataset and training setup. This need for customization adds complexity to the implementation.
>
> Thanks for your valuable feedback.
>
> * First, regarding the parameters for token retention rates, we simply **follow the experimental setup of other recognized token pruning studies**, such as EViT and ToMe, and **report accuracy and computation cost across various token keep rates.** It is worth emphasizing that SimPrune consistently outperforms these prior works under all tested token retention rates. Token retention rate is a widely adopted user-defined hyper-parameter in token pruning methods, typically configured to achieve the desired level of computation savings based on application requirements.
>
>
> We have also shown that our approaches are complementary and can be easily extended to the dynamic pruning strategy in Question Q2 and **Table R.1**. **Please refer to Question Q2 for more details.**
>
>
> * Second, for the parameters of pruning stages, we want to point out that the training stages are automatically defined based on the training epochs, without any manual effort. Once the training epochs are appropriately set, the sliding window just linearly moves from pruning the most dissimilar token pairs to the most similar tokens according to the training epochs.
>
>
> ---
>
>
>
> ### Q6. While SimPrune seeks to maintain semantic consistency through cross-branch pruning, low precision in token matching may still cause semantic inconsistencies between branches, which can negatively impact model performance.
>
> Thank you for your insightful feedback. In SimPrune, token similarity is measured using cosine similarity, a widely used and well-established metric in related research areas that has been proven effective [2]. To ensure cross-branch semantic consistency, SimPrune matches tokens with the highest cosine similarity across the two branches, aligning semantically similar tokens.
>
> As shown in our visualization in Figure 4 and detailed in Section 5.4, **SimPrune produces pruning regions that are more concentrated and symmetrical** compared to attention-based token pruning methods, highlighting its ability to maintain semantic consistency through this matching technique. Furthermore, our experimental results validate the effectiveness of SimPrune in achieving superior performance.
>
>
> ---
>
>
>
> ### Q7. During initial training, SimPrune’s intensive token matching and pruning calculations lead to high resource demands. While these requirements lessen in later stages, the early computational load may pose challenges for resource-limited devices or environments.
>
> Thanks for your feedback on the overhead. We want to point out that we keep the same prune ratio across the whole training process. The extra overhead for token matching is **less than 1%** compared to the total cost. The memory usage for token matching is also negligible. This is because the token similarity is measured by cosine similarity, which is pretty lightweight in both computation and memory usage.

---

> ### Author Response · Authors · 2024-12-02
> **Author Response to Reviewer 1ySw**
>
> Dear Reviewer 1ySw,
>
> Thanks for your time and reviewing efforts! We appreciate your constructive comments.
>
> We have provided suggested results in the authors' response, such as the comparison with more SOTA work and extending our proposed SimPrune to incorporate a dynamic pruning strategy. We have also provided clarification and explanation regarding your concerns such as hyper-parameters tuning.
>
> We hope our responses have answered your questions. It would be our great pleasure if you would consider updating your review or score.
>
> Best,
>
> Authors

---

### Meta-Review · Area_Chair_3oFw · 2024-12-17

**Metareview:**

This paper proposes a new token pruning technique, SimPrune, for dual-branch SSL for more efficient training. It leverages cross-branch similarity information to efficiently prune tokens, retaining essential semantic information across dual branches. In addition, it uses a dynamic pruning strategy, where the early/late stage prunes the low/high similarity tokens. Experiments show that SimPrune can reduce training costs by~24% while maintaining accuracy.

All reviewers are unanimously positive on this submission, and like the contributions including 1) the proposed techniques are reasonable and well motivated; 2) the experiments support the proposed claims; 3) the paper is well written.The common concerns from the reviewers are 1) improvement is incremental on performance/cost; 2) computation overhead; 3) missing experiments on dense tasks, larger models. However, most of the concerns were well addressed after the rebuttal. The AC agrees with the reviewers' unanimous decision on accept.

**Additional Comments On Reviewer Discussion:**

For 1ySw, the concerns are: the results are inferior; more computations; sensitivity on HPs; higher training difficulty; potential on semantic inconsistency. For 8Qhp, the concerns are: missing reference, missing experiments on dense tasks; unclear motivation, missing experiments on large models. For JVJ6, the concerns are: missing reference; missing experiments on dense tasks; missing some clarification. For aES2,  the concerns are: results difference from the baseline models; unclear motivation on performance/speed.

The authors did a good rebuttal by providing more experimental results, missing reference/ablations, and all reviewers are happy with this submission in the end. 1ySw, 8Qhp and aES2 increased the score.

---

### Decision · Program_Chairs · 2025-01-22

Accept (Poster)